# PREFERENCE ORCHESTRATOR: PROMPT-AWARE MULTI-OBJECTIVE ALIGNMENT FOR LARGE LANGUAGE MODELS

## ABSTRACT

While Large Language Models (LLMs) have demonstrated remarkable capabilities across diverse natural language processing tasks, aligning these models with varying human preferences across multiple objectives remains a significant challenge in practical deployments. Existing multi-objective alignment methods rely on manually specified preference weights, which not only burden users with difficult preference specification tasks but also lead to suboptimal training efficiency due to exploration of irrelevant preference combinations. To alleviate these issues, we propose a novel framework named PRO, i.e., PReference Orchestrator, which features a lightweight preference adapter that automatically infers prompt-specific preference weights during both training and deployment phases. Specifically, the adapter automatically learns appropriate preference weights for each prompt by training on normalized reward scores from multiple reward models for preferred responses, which inherently reflect effective preference balances across objectives. Additionally, We provide theoretical analysis proving that our prompt-aware preference mechanism achieves superior performance compared to fixed preference weights in multi-objective alignment scenarios. Extensive experiments across multiple tasks demonstrate the effectiveness of our method over existing multi-objective alignment approaches.

## 1 INTRODUCTION

Large Language Models (LLMs) have demonstrated remarkable capabilities across a wide range of natural language processing tasks, including text generation (Liang et al., 2024), conversational interaction (Wang et al., 2023), reasoning (Xu et al., 2025), and code completion (Jiang et al., 2024). However, ensuring that these models align with human values and preferences remains a significant challenge. Misaligned models can produce outputs that are biased, harmful, or harmless but unhelpful, leading to negative user experiences and potential societal harm. Therefore, effective alignment techniques are crucial for deploying LLMs in real-world applications, with RLHF, i.e., Reinforcement Learning from Human Feedback (Ziegler et al., 2019; Stiennon et al., 2020; Ouyang et al., 2022), being one of the most prominent methods.

In practical deployments, different users often have diverse preferences regarding LLM outputs. For instance, some may prioritize helpfulness and informativeness, while others might value safety and harmlessness more highly. A single objective is insufficient to capture these multi-dimensional requirements. Multi-objective alignment aims to train models that can adapt to these varying preference profiles, typically represented as a preference weight vector, where each dimension corresponds to the relative importance of a particular objective (Li et al., 2021; Rame et al., 2023; Yang et al., 2024b).

A straightforward approach for multi-objective alignment is to combine multiple reward models into a single reward signal through weighted summation, then use the combined reward signal for RL optimization (Li et al., 2021). While effective, this approach typically uses fixed weights during training, and developing separate models for different preference combinations remains resource-intensive. To address the inefficiency, methods like MODPO (Zhou et al., 2024b) and CPO (Guo et al., 2024) have eliminated the RL step entirely by optimizing directly on multi-objective preference data.

More recent innovations focus on test-time adaptability to user preferences (Rame et al., 2023; Yang et al., 2024b), enabling a single model to accommodate diverse preference profiles. For instance, REWARD SOUPS (Rame et al., 2023) and MOD (Shi et al., 2024) train multiple single-objective expert models for each objective, then perform weighted averaging of these experts based on user preferences at test time. RIC (Yang et al., 2024b) and DPA (Wang et al., 2024) control user preferences by appending reward scores to the input, followed by SFT fine-tuning. During online sampling, they randomly sample user preferences to generate new responses and use rejection sampling (Dong et al., 2023) to filter high-quality samples for further iterative training. PARM (Lin et al., 2025) employs a single preference-aware autoregressive reward model that dynamically adapts to user-specified preference vectors to guide a frozen base model's generation process.

However, while existing methods can adapt to different preferences for each prompt, they rely on manually specified preference weights. In practice, users often struggle to determine the optimal preference combination for a given prompt—for instance, how to properly balance honesty, helpfulness, and harmlessness when asking for advice about a sensitive political topic. This dependency on manual input for preference weights not only increases user burden but may also lead to suboptimal output quality due to inappropriately preferences setting. Additionally, during the training phase, approaches like RIC and DPA employ random sampling of preference vectors to increase training data diversity, but these randomly sampled preferences may deviate from the optimal configuration for specific prompts. This results in reduced training efficiency and computational resources wasted exploring ineffective preference combinations. To address these limitations, we propose Prompt-Aware Multi-Objective Alignment with a *Preference Orchestrator* that automatically infers appropriate preference weight vectors for each prompt, eliminating the need for manual input while providing more intelligent preference sampling strategies during training.

Motivated by the above consideration, we introduce a novel framework named PRO, i.e., *PReference Orchestrator*, which involves a lightweight adapter module that automatically learns appropriate preference weights for multi-objective alignment. Specifically, the adapter takes an input prompt and outputs a weight vector that specifies how to combine multiple reward objectives for that specific context. The adapter is trained on normalized reward scores from multiple reward models for the preferred responses in existing human preference data, leveraging the insight that preferred responses inherently reflect effective preference balances across objectives. Additionally, our framework serve as a flexible plugin that can be integrated with existing multi-objective alignment methods, enhancing their performance by providing prompt-aware preference rather than relying on random sampling or fixed weights. Our contributions are summarized as follows:

- **Practically**, we propose the PRO framework, a lightweight and flexible preference adapter that automatically infers preference weights without requiring manual specification. This framework can be seamlessly integrated with existing multi-objective alignment methods as a plug-in module, enhancing their performance while reducing user burden and improving training efficiency.

- **Theoretically**, we prove that our prompt-aware preference mechanism achieves superior performance compared to using fixed preference weights, providing theoretical guarantees for the effectiveness of adaptive preference in multi-objective alignment scenarios.

Extensive experiments on multiple tasks, including summarization, question answering, and mathematical reasoning, demonstrate the effectiveness of our method over existing multi-objective alignment approaches.

## 2 RELATED WORK

**Language Model Alignment**: Aligning LLMs with human values and intentions is a fundamental step toward building responsible and effective AI systems (Achiam et al., 2023; Chen et al., 2025). The most influential paradigm is Reinforcement Learning with Human Feedback (RLHF) (Christiano et al., 2017; Stiennon et al., 2020; Ouyang et al., 2022), where a reward model is first trained to capture human preference signals, and the LLM is subsequently fine-tuned to maximize the expected reward under a KL-regularized objective. Despite its effectiveness, RLHF suffers from high computational cost and training instability (Dong et al., 2023; Yuan et al., 2023). To address these issues, DPO (Rafailov et al., 2023) was proposed as a simpler and more efficient alternative. DPO directly learns from pairwise human preference data and has been shown to be mathematically

equivalent to RLHF under certain assumptions. This perspective has inspired a series of variants aiming to further improve optimization efficiency, stability and alignment quality (Ethayarajh et al., 2024; Hong et al., 2024; Meng et al., 2024; Kim et al., 2025; Garg et al., 2025). For instance, SIMPO (Meng et al., 2024) eliminates the dependency on a reference model and mitigates length bias in optimization by introducing a length regularization term, resulting in more efficient training. KTO (Ethayarajh et al., 2024) proposes a divergence-based formulation that directly operates on binary feedback, thereby avoiding the need for pairwise preference comparisons while maintaining stable alignment.

**Multi-Objective Language Model Alignment**: Multi-objective alignment aims to optimize language models across multiple, potentially conflicting objectives such as helpfulness, harmlessness, and honesty. Early approaches typically employ weighted summation to combine multiple reward models into a unified signal for reinforcement learning optimization (Li et al., 2021). However, these methods rely on fixed preference weights throughout training, limiting their adaptability to diverse user needs and requiring separate models for different preference combinations. Recent work has explored more efficient alternatives that eliminate the computationally expensive RL step. Methods like MODPO (Zhou et al., 2024b) and CPO (Guo et al., 2024) directly optimize on multi-objective preference data, avoiding the instability and computational overhead associated with RL-based approaches. A growing line of research focuses on runtime adaptability, enabling a single model to accommodate diverse user preferences (Rame et al., 2023; Yang et al., 2024b). REWARD SOUPS (Rame et al., 2023) and MOD (Shi et al., 2024) train multiple single-objective expert models and perform weighted averaging at inference time based on user-specified preferences. RIC (Yang et al., 2024b) and DPA (Wang et al., 2024) control preferences by appending reward scores to inputs during supervised fine-tuning, then use rejection sampling (Dong et al., 2023) during inference to filter high-quality responses. PARM (Lin et al., 2025) employs a preference-aware autoregressive reward model that dynamically adapts to user-specified preference vectors to guide generation from a frozen base model. While these approaches demonstrate promising results, they either require training multiple specialized models or rely on explicit user preference specification at inference time.

## 3 PRELIMINARIES

We first introduce the formal notation for the language model alignment with single reward model. Let $\mathcal{V}$ be a vocabulary of a language model. The goal of alignment is to ensure that the language model $\pi : \mathcal{X} \to \mathcal{Y}$ generates response $\boldsymbol{y} \in \mathcal{Y}$ that are consistent with human values and preferences given a query $\boldsymbol{x} \in \mathcal{X}$, where the query $\boldsymbol{x} = [x^1, x^2, \ldots, x^m]$ and response $\boldsymbol{y} = [y^1, y^2, \ldots, y^n]$ are sequences of tokens, the input space $\mathcal{X} = \mathcal{V}^m$ and the output space $\mathcal{Y} = \mathcal{V}^n$.

**Supervised Fine-Tuning (SFT)**: The alignment process typically begins with Supervised Fine-Tuning (SFT), which adjusts the language model using Maximum Likelihood Estimation on a human-labeled high-quality dataset $\mathcal{D}_{\text{sft}} = \{(\boldsymbol{x}_i, \boldsymbol{y}_i)\}_{i=1}^N$:

$$\mathcal{L}_{\text{SFT}} = -\sum_{i=1}^{N} \sum_{j=1}^{n_i} \log P(y_i^j | [y_i^k]_{k=0}^{j-1}, \boldsymbol{x}^{(i)}; \theta), \tag{1}$$

where $N$ is the number of training examples, $n_i$ is the length of the $i$-th target sequence, and $\theta$ represents the parameters of the language model $\pi_\theta$. For the notational simplicity, $y_i^0 = \emptyset$ denotes an empty placeholder.

**Reinforcement Learning from Human Feedback (RLHF)**: To further align the language model with human preferences, Reinforcement Learning from Human Feedback (RLHF) is employed. This involves training a reward model $r_\phi : \mathcal{X} \times \mathcal{Y} \to \mathbb{R}$ using a dataset of human preferences $\mathcal{D}_{\text{rm}} = \{(\boldsymbol{x}_i, \boldsymbol{y}_i^+, \boldsymbol{y}_i^-)\}_{i=1}^M$, where each entry consists of a query $\boldsymbol{x}_i$ and two responses $\boldsymbol{y}_i^+$ and $\boldsymbol{y}_i^-$, with $\boldsymbol{y}_i^+$ being preferred over $\boldsymbol{y}_i^-$. The reward model is trained to satisfy the following condition:

$$\mathcal{L}_{\text{rm}} = -\sum_{i=1}^{M} \log P(\boldsymbol{y}_i^+ \succ \boldsymbol{y}_i^- | \boldsymbol{x}_i; \phi) = -\sum_{i=1}^{M} \log \sigma(r_\phi(\boldsymbol{x}_i, \boldsymbol{y}_i^+) - r_\phi(\boldsymbol{x}_i, \boldsymbol{y}_i^-)), \tag{2}$$

where $\sigma(\cdot)$ is the sigmoid function. Subsequently, the language model is fine-tuned using reinforcement learning algorithms, such as Proximal Policy Optimization (PPO) (Schulman et al., 2017), to

maximize the expected reward provided by the reward model:

$$\mathcal{L}_{\text{RLHF}}(\theta) = \mathbb{E}_{\boldsymbol{x} \sim \mathcal{D}, \, \boldsymbol{y} \sim \pi_\theta(\cdot|\boldsymbol{x})} \left[ -r_\phi(\boldsymbol{x}, \boldsymbol{y}) + \beta \operatorname{KL}(\pi_\theta(\cdot|\boldsymbol{x}) \,\|\, \pi_{\text{ref}}(\cdot|\boldsymbol{x})) \right], \tag{3}$$

where $\beta$ is a hyperparameter that balances the reward maximization and the Kullback-Leibler (KL) divergence regularization term, which prevents the fine-tuned model from deviating excessively from the reference model $\pi_{\text{ref}}$, which is typically the SFT model.

**Multi-Objective Alignment**: In practical scenarios, aligning a language model with multiple, often conflicting, human preferences is essential. This is typically achieved by training multiple reward models $\{r_{\phi_k}\}_{k=1}^K$ with the multi-objective dataset $\mathcal{D}_{\text{mo}} = \{(\boldsymbol{x}_i, \boldsymbol{y}_{i1}, \boldsymbol{y}_{i2}, \{p_{i,k}\}_{k=1}^K)\}_{i=1}^M$, where $p_{i,k} \in \{0, 1\}$ denotes the preference for the $k$-th objective. $p_{i,k} = 1$ indicates that response $\boldsymbol{y}_{i1}$ is preferred over $\boldsymbol{y}_{i2}$ for the $k$-th objective, and vice versa. The typical approach involves combining these reward models into a single scalar reward using a weighted sum:

$$r_{\text{mo}}(\boldsymbol{x}, \boldsymbol{y}; \boldsymbol{w}) = \sum_{k=1}^K w_k r_{\phi_k}(\boldsymbol{x}, \boldsymbol{y}), \tag{4}$$

where $\boldsymbol{w} = [w_1, w_2, \ldots, w_K]$ are non-negative weights that sum to one, reflecting the relative importance of each objective. The language model is then fine-tuned using the combined reward in a manner similar to Eq. (3):

$$\mathcal{L}_{\text{MORLHF}}(\theta; \boldsymbol{w}) = \mathbb{E}_{\boldsymbol{x} \sim \mathcal{D}, \, \boldsymbol{y} \sim \pi_\theta(\cdot|\boldsymbol{x})} \left[ -r_{\text{mo}}(\boldsymbol{x}, \boldsymbol{y}; \boldsymbol{w}) + \beta \operatorname{KL}(\pi_\theta(\cdot|\boldsymbol{x}) \,\|\, \pi_{\text{ref}}(\cdot|\boldsymbol{x})) \right]. \tag{5}$$

**Test-Time Multi-Objective Alignment**: At test time, users may have different preferences for the importance of each objective. To accommodate this, the language model can be adapted to user-specified weights $\boldsymbol{w}$ without retraining. Formally, the response of each prompt $\pi(\boldsymbol{y}|\boldsymbol{x}, \boldsymbol{w})$ is conditioned on both the input prompt $\boldsymbol{x}$ and the preference weights $\boldsymbol{w}$.

# 4 THE PROPOSED METHOD

## 4.1 THE PREFERENCE ORCHESTRATOR

In this section, we introduce the PRO, i.e., PREFERENCE ORCHESTRATOR, a lightweight classifier module that automatically determines the optimal preference weight vector for multi-objective alignment given an input prompt. The adapter takes an input prompt $\boldsymbol{x}$ and outputs a weight vector $\boldsymbol{w} = [w_1, w_2, \ldots, w_K]$ that specifies how to combine multiple reward objectives for that specific context. This learned adapter enables prompt-aware optimization, where different types of inputs can be automatically assigned appropriate preference configurations based on their characteristics. Formally, we define the adapter as $\boldsymbol{w} = f_\psi(\boldsymbol{x})$, where $f_\psi : \mathcal{X} \rightarrow \Delta^{K-1}$ is a neural network parameterized by $\psi$, and $\Delta^{K-1}$ represents the $(K-1)$-simplex ensuring valid probability distributions with $\sum_{k=1}^K w_k = 1$ and $w_k \geq 0$.

## 4.2 TRAINING THE PREFERENCE ORCHESTRATOR

To train the *Preference Orchestrator*, we leverage the existing preference dataset $\mathcal{D}_{\text{rm}} = \{(\boldsymbol{x}_i, \boldsymbol{y}_i^+, \boldsymbol{y}_i^-)\}_{i=1}^M$. The key insight is that the preferred responses inherently reflect an effective balance across multiple objectives—they are preferred precisely because they achieve a superior trade-off among various quality dimensions. For instance, a technical query might yield a preferred response with high scores on accuracy and informativeness, while a creative writing prompt might have preferred responses scoring highly on creativity and engagement. These score distributions implicitly encode the context-appropriate preference weights.

To extract the implicit preference weights from these preferred responses, we compute the rewards from all $K$ reward models for each preferred response:

$$\boldsymbol{r}_i^+ = [r_{\phi_1}(\boldsymbol{x}_i, \boldsymbol{y}_i^+), r_{\phi_2}(\boldsymbol{x}_i, \boldsymbol{y}_i^+), \ldots, r_{\phi_K}(\boldsymbol{x}_i, \boldsymbol{y}_i^+)]. \tag{6}$$

We then normalize these reward scores to obtain valid preference weights:

$$\boldsymbol{w}_i^* = \operatorname{softmax}(\boldsymbol{r}_i^+/\tau) = \left[ \frac{\exp(r_{\phi_k}(\boldsymbol{x}_i, \boldsymbol{y}_i^+)/\tau)}{\sum_{j=1}^K \exp(r_{\phi_j}(\boldsymbol{x}_i, \boldsymbol{y}_i^+)/\tau)} \right]_{k=1}^K, \tag{7}$$

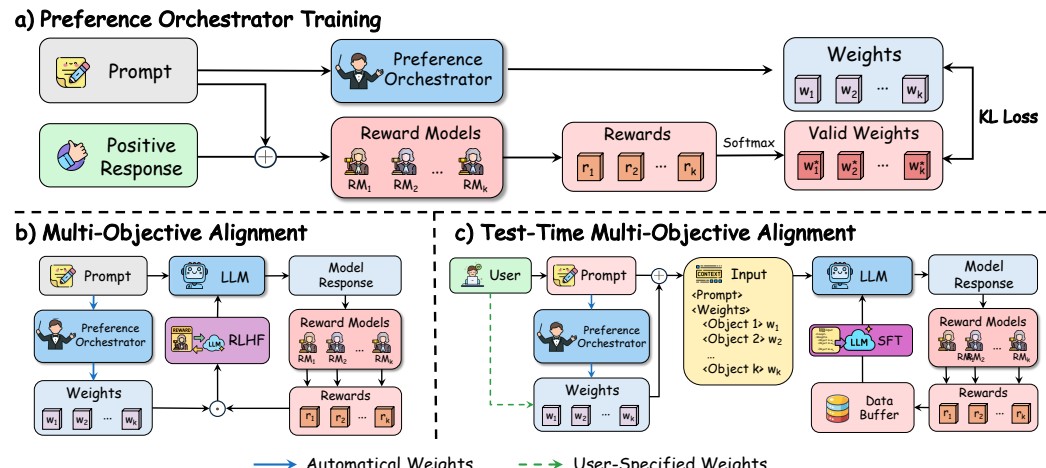

Figure 1: Overview of the PRO framework. The adapter takes an input prompt and outputs a weight vector that determines how to combine multiple reward objectives for that specific context.

where $\tau$ is a temperature parameter that controls the sharpness of the distribution.

The is then trained using supervised learning to predict these implicit preference weights:

$$\mathcal{L}_{\text{PRO}}(\psi) = \frac{1}{M} \sum_{i=1}^{M} \text{KL}(f_\psi(\boldsymbol{x}_i) \| \boldsymbol{w}_i^*), \tag{8}$$

where KL denotes the Kullback-Leibler divergence between the predicted and target weight distributions. This training objective enables the adapter to learn the mapping from prompt characteristics to optimal preference configurations, distilling the implicit preferences encoded in human-annotated data into an explicit weight prediction mechanism. The training of PRO is illustrated in Figure 1 (a).

### 4.3 INTEGRATING THE PREFERENCE ORCHESTRATOR WITH MULTI-OBJECTIVE ALIGNMENT

The PRO can be seamlessly integrated into existing multi-objective alignment frameworks. During both training and inference, the adapter generates context-specific preference weights for each input prompt, which are then used to combine the multiple reward models.

**Integrating with Multi-Objective Alignment**: In the multi-objective alignment setting, where users input only prompt without any explicit preference weights, we utilize the PRO to generate weights for each prompt during the training phase, making the model implicitly learn the ability to generate responses that trade off between multiple objectives. Taking MORLHF as an example:

$$\mathcal{L}_{\text{PRO-MORLHF}}(\theta; f_\psi) = \mathbb{E}_{\boldsymbol{x} \sim \mathcal{D}, \boldsymbol{y} \sim \pi_\theta(\cdot | \boldsymbol{x})} \left[ -r_{\text{mo}}(\boldsymbol{x}, \boldsymbol{y}; f_\psi(\boldsymbol{x})) + \beta \, \text{KL}(\pi_\theta(\cdot | \boldsymbol{x}) \| \pi_{\text{ref}}(\cdot | \boldsymbol{x})) \right]. \tag{9}$$

This approach allows the model to adaptively focus on the most relevant objectives for each prompt, leading to more effective and contextually appropriate responses. The architecture of this integration is illustrated in Figure 1 (b).

**Integrating with Test-Time Multi-Objective Alignment**: In the Test-Time Multi-Objective Alignment setting, our method provides dual advantages. First, during the online sampling phase, our approach can provide recommended preference configurations, avoiding potentially unreasonable preference combinations that may arise from random sampling, thereby improving training efficiency and reducing computational resource waste. Second, during inference, when users do not have explicit preference specifications, our method can automatically provide reasonable default preference weights, ensuring consistency in model output quality. Motivated by the reward in context technique (Lu et al., 2022; Yang et al., 2024b; Wang et al., 2024), we encode the preference weights as additional input tokens appended to the original prompt. The integration process involves two stages:

**Offline Stage**: During the offline training phase, the model is first warmed up using weights-conditioned supervised fine-tuning. For each training sample $(\boldsymbol{x}_i, \boldsymbol{y}_i)$, we first compute the rewards from all $K$ reward models and normalize them using softmax to obtain preference weights by Eq. (7). The offline training objective becomes:

$$\mathcal{L}_{\text{PRO-WIC}}^{\text{offline}}(\theta) = -\sum_{i=1}^{N}\sum_{j=1}^{n_i} \log P(y_i^j | [y_i^k]_{k=0}^{j-1}, \boldsymbol{x}_i, \boldsymbol{w}^*; \theta), \tag{10}$$

where the input $\boldsymbol{x}_i, \boldsymbol{w}^*$ is constructed by appending these normalized weights to the original prompt by the template: `Prompt <W1> `$w_{i,1}^*$` <W2> `$w_{i,2}^*$` ... <WK> `$w_{i,K}^*$. This stage serves as a warm-up phase, teaching the model to respond conditioned on preference weights.

**Online Sampling Stage**: During the online phase, our adapter recommends preference weights, replacing the random preference sampling strategy used in previous methods (Yang et al., 2024b; Wang et al., 2024).

$$\mathcal{L}_{\text{PRO-WIC}}^{\text{online}}(\theta; f_\psi) = -\sum_{\boldsymbol{x}_i \in \mathcal{D}_{\text{online}}}\sum_{j=1}^{n_i} \log P(y_i^j | [y_i^k]_{k=0}^{j-1}, \boldsymbol{x}_i, f_\psi(\boldsymbol{x}_i); \theta), \tag{11}$$

where $\mathcal{D}_{\text{online}} = \{\boldsymbol{x}_i\}_{i=1}^{O}$ is the online prompt set, and $f_\psi(\boldsymbol{x}_i)$ provides the adapter-predicted preference weights for prompt $\boldsymbol{x}_i$. The architecture of this integration is illustrated in Figure 1 (c).

This adaptive mechanism enables our framework to both satisfy users with explicit preferences and provide intelligent solutions for scenarios lacking preference guidance, making the system more user-friendly and practically deployable.

## 5 THEOREMTICAL ANALYSIS

In this section, we provide a theoretical analysis of the *Preference Orchestrator* and its impact on multi-objective alignment. We consider two approaches for multi-objective alignment:

- **Fixed-weight approach**: Uses a single global weight vector $\boldsymbol{w}_{\text{fixed}} \in \mathcal{W}$ for all prompts, typically set as uniform weights $\boldsymbol{w}_{\text{fixed}} = [1/K, \ldots, 1/K]$.
- **Adaptive approach**: Uses our *Preference Orchestrator* $f_\psi : \mathcal{X} \to \Delta^{K-1}$ to generate context-specific weights for each prompt.

For a given prompt $\boldsymbol{x}$, the alignment gap measures the suboptimality of a policy $\pi$ compared to the optimal policy $\pi_{\boldsymbol{w}^*(\boldsymbol{x})}^*$ under the true optimal weights $\boldsymbol{w}^*(\boldsymbol{x})$ is defined as:

$$\text{Gap}(\pi, \boldsymbol{x}) = F_{r_{\text{mo}}(\cdot; \boldsymbol{w}^*(\boldsymbol{x}))}(\pi_{\boldsymbol{w}^*(\boldsymbol{x})}^*) - F_{r_{\text{mo}}(\cdot; \boldsymbol{w}^*(\boldsymbol{x}))}(\pi), \tag{12}$$

where $F_r(\pi) = \mathbb{E}_{\boldsymbol{y} \sim \pi(\cdot|\boldsymbol{x})}[r(\boldsymbol{x}, \boldsymbol{y})] - \beta D_{\text{KL}}[\pi(\cdot|\boldsymbol{x}) \| \pi_{\text{ref}}(\cdot|\boldsymbol{x})]$ is the KL-regularized reward objective and $\pi_{\boldsymbol{w}^*(\boldsymbol{x})}^* = \min_{\pi \sim \mathcal{H}} F_{r_{\text{mo}}(\cdot; \boldsymbol{w}^*(\boldsymbol{x}))}(\pi)$, $\mathcal{H}$ is the hypothesis space.

The overall alignment gap is then defined as the expected gap over the prompt distribution:

$$\text{Align-Gap}(\pi) = \mathbb{E}_{\boldsymbol{x} \sim \mathcal{D}}[\text{Gap}(\pi, \boldsymbol{x})]. \tag{13}$$

We now present our main theoretical result, which demonstrates that the adaptive weight approach using the *Preference Orchestrator* achieves a smaller lower bound of alignment gap compared to the fixed-weight approach.

**Theorem 5.1** (Superiority of Adaptive Weights). *Let $\pi_{fixed}$ be the optimal policy trained with fixed weights $\boldsymbol{w}_{fixed}$, and $\pi_{adapt}$ be the policy optimized using our Preference Orchestrator $f_\psi$. Under the following assumptions: (i) The reward function $r_{mo}(\cdot; \boldsymbol{w})$ is Bi-Lipschitz continuous lower bouned by $L_r$ with respect to the weight vector $\boldsymbol{w}$; (ii) The KL-regularized objective satisfies strong convexity with parameter $\mu > 0$; (iii) The reward objective $F_r(\pi)$ is lower bounded by a constant $C > 0$, i.e., $\min_{\pi, r, \boldsymbol{w}} F_{r_{mo}(\cdot; \boldsymbol{w})}(\pi) = C$; then the alignment gaps satisfy:*

$$\text{Align-Gap}(\pi_{fixed}) \geq \frac{\mu L_r^2}{2\beta^2 L_\pi^2} \mathbb{E}_{\boldsymbol{x} \sim \mathcal{D}}\left[\|\boldsymbol{w}^*(\boldsymbol{x}) - \boldsymbol{w}_{fixed}\|_2^2\right]$$

$$\text{Align-Gap}(\pi_{adapt}) \geq \frac{\mu L_r^2 C^2}{2\beta^2 L_\pi^2} \mathcal{O}\left(\frac{\log \frac{1}{\delta}}{N}\right). \tag{14}$$

with probability at least $1 - \delta$, where $N$ is the number of training samples of the Preference Orchestrator and $L_\pi$ is the Lipschitz constant of the log-policy function. The proof is provided in Appendix A.1.

**Remark.** *Theorem 5.1 reveals the advantage of our adaptive approach over fixed-weight methods. As the number of training samples $N$ approaches infinity, the alignment gap of our Preference Orchestrator approaches zero, indicating that our method can achieve near-optimal performance with sufficient training data. In contrast, the fixed-weight approach maintains a persistent lower bound on its alignment gap that is proportional to $\mathbb{E}_{\boldsymbol{x} \sim \mathcal{D}}[\|\boldsymbol{w}^*(\boldsymbol{x}) - \boldsymbol{w}_{fixed}\|_2^2]$, representing the inherent mismatch between the global fixed weights and the context-specific optimal weights. This fixed error becomes increasingly problematic as the diversity of optimal preferences across different prompts grows larger, highlighting the limitation of using uniform weights for all contexts.*

# 6 EXPERIMENTS

## 6.1 EXPERIMENTAL SETUP

**Datasets and Models.** For test-time multi-objective alignment setting, we evaluated our approach on two datasets: Reddit Summary (Völske et al., 2017) and Helpful Assistant (Bai et al., 2022). The Reddit Summary dataset contains summaries of Reddit posts, comprising 14.9k posts and corresponding summaries. We consider reward models: preference and summaries, which evaluate human preference for summaries trained with different datasets, and a faithful reward that measures the faithfulness of the summary to the original post. Helpful Assistant is a dialogue task containing 160k prompts and corresponding responses, annotated with human preferences. We employ three reward models for this dataset: helpfulness, harmlessness, and humor. For multi-objective alignment setting, we evaluated our approach on Ultrafeedback (Cui et al., 2023), which is a fine-grained, diverse preference dataset with 64k prompts and corresponding responses, annotated with human preferences across four dimensions: instruction-following, truthfulness, honesty, and helpfulness. We trained separate reward models for each of these dimensions. For Reddit Summary and Helpful Assistant, we used LLaMA-7B (Touvron et al., 2023) as the base model, while for Ultrafeedback, we employed Qwen-2.5-7B (Yang et al., 2024a) as the base model.

**Evaluation Metrics.** For Reddit Summary and Helpful Assistant, we randomly sampled 2k prompts from the test set, generated responses with different weights of user preferences, and calculated the average score for each reward dimension. We compared the multi-dimensional average test reward curves corresponding to the empirical Pareto frontiers generated by different methods. The outer curves indicate superior performance of the method across objectives with various preferences. For Ultrafeedback, we employed three widely adopted automatic evaluation benchmarks for LLMs: AlpacaEval 2 (Li et al., 2023; Dubois et al., 2024), MT-Bench (Bai et al., 2024), and Arena-Hard (Li et al., 2024a;b). All evaluations used GPT-4o as the judge model. For AlpacaEval 2, we report the raw win rate (WR) and length-controlled win rate (LC) against the reference model GPT-4o-05-13. For Arena-Hard, we report the win rate (WR) and style-controlled win rate (SC), comparing our model against the GPT-4-Preview-1106 baseline. For MT-Bench, we report the average multi-turn score (Score) assigned by GPT-4o, which rates each response on a scale of 1-10.

**Baselines.** We compared our approach with two different types of baseline methods. For Reddit Summary and Helpful Assistant datasets, we compared with multi-objective alignment methods including: (1) MORLHF (Li et al., 2021): This method assigns fixed weights to each objective reward model, using the weighted score as the reward signal for PPO optimization. (2) REWARD SOUPS (Rame et al., 2023): This approach first trains multiple expert models using single-objective RL, then performs weighted averaging of these experts' outputs, with weights determined by user preferences. (3) RIC (Yang et al., 2024b): This method appends reward scores to the prompt to control user preferences, followed by fine-tuning using SFT. For Ultrafeedback, we compared our method with various advanced LLM alignment methods: SFT, DPO (Rafailov et al., 2023), IPO (Azar et al., 2024), KTO (Ethayarajh et al., 2024), SIMPO (Meng et al., 2024), WPO (Zhou et al., 2024a), SELECTIVE DPO (Gao et al., 2025), ADPO (Ji et al., 2025), and PPO (Ouyang et al., 2022). The implementation details are provided in Appendix A.2.

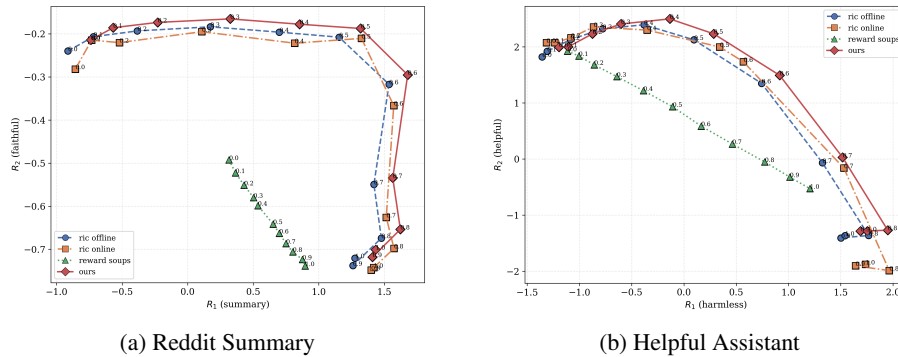

|                      |                      |
| :------------------: | :------------------: |
| (a) Reddit Summary   | (b) Helpful Assistant |

Figure 2: Results of the Reddit Summary and Helpful Assistant in test-time multi-objective alignment.

Table 1: Performance on Reddit Summary with two objectives (Equal weights).

| Method        | Summary | Faithful |
| :------------ | :------ | :------- |
| MORLHF        | 0.78    | -0.66    |
| REWARD SOUPS  | 0.65    | -0.64    |
| RIC offline   | 1.15    | -0.21    |
| RIC online    | 1.35    | -0.21    |
| PRO-WIC       | **1.46** | **-0.19** |

Table 2: Performance on Reddit Summary with three objectives (Equal weights).

| Method        | Summary | Faithful | Preference |
| :------------ | :------ | :------- | :--------- |
| MORLHF        | 0.78    | -0.66    | 0.55       |
| REWARD SOUPS  | 0.64    | -0.56    | 0.91       |
| RIC offline   | 0.71    | -0.25    | 1.33       |
| RIC online    | 0.84    | -0.25    | 1.69       |
| PRO-WIC       | **0.95** | **-0.23** | **2.12**  |

## 6.2 MAIN RESULTS

**Performance on Reddit Summary and Helpful Assistant Tasks.** As shown in Figure 2, each point in the figure represents the average score across all reward dimensions. The numbers at the centers of the markers indicate the preference weight for the first reward in each pair. Due to the substantial computational cost of MORLHF for various preference weight combinations and the inability to adapt to different user preferences in test-time, we do not report the results for MORLHF in the figure. Compared to baseline methods, the curve for our method, i.e., PRO-WIC, consistently lies on the outermost boundary in most cases, indicating that our method can adapt to different user preferences and balance multiple conflicting objectives. Furthermore, we compare the performance of different methods under an equal-weight setting. As shown in Tables 1-4, PRO achieves the best scores on most evaluation metrics in both two-objective and three-objective scenarios (except for the Helpful dimension on Helpful Assistant).

**General Capability Assessment on Ultrafeedback.** To evaluate the general capabilities of our method in broader scenarios, we trained it on the Ultrafeedback dataset and tested it on three mainstream benchmarks: AlpacaEval 2, Arena-Hard, and MT-Bench. As shown in Table 5, PRO outperforms almost all baseline methods across multiple benchmarks. Specifically, on AlpacaEval 2, PRO-MORLHF achieves a win rate (WR) and length-controlled win rate (LC) of 47.30% and 50.35%, respectively, significantly outperforming all baselines. On the more challenging Arena-Hard benchmark, our method also demonstrates strong competitiveness. On MT-Bench, PRO-MORLHF achieves the score of 7.93, which is only slightly lower than the best baseline ADPO.

## 6.3 ABLATION STUDY

**PRO-WIC vs. RIC Variants.** In the test-time multi-objective alignment setting, we compare our method PRO-WIC with two variants of RIC: RIC offline (which removes the online sampling phase) and RIC online (which uses random preference sampling during the online phase). As shown in Tables 1-4, PRO-WIC consistently outperforms both RIC variants across all evaluation scenarios.

**PRO-MORLHF vs. MORLHF.** In the multi-objective alignment setting, we compare our method PRO-MORLHF with the baseline MORLHF approach. As shown in Table 5, MORLHF uses fixed uniform weights across all prompts, while our method employs the *Preference Orchestrator* to assign

Table 3: Performance on Helpful Assistant with two objectives (Equal weights).

| Method | Harmless | Helpful |
|--------|----------|---------|
| MORLHF | 0.31 | 0.76 |
| REWARD SOUPS | -0.11 | 0.93 |
| RIC offline | 0.10 | 1.86 |
| RIC online | 0.34 | 2.00 |
| PRO-WIC | **0.57** | **2.10** |

Table 4: Performance on Helpful Assistant with three objectives (Equal weights).

| Method | Harmless | Helpful | Humor |
|--------|----------|---------|-------|
| MORLHF | 0.31 | 0.76 | -0.35 |
| REWARD SOUPS | 0.02 | 0.66 | 0.39 |
| RIC offline | -0.51 | 1.22 | 0.82 |
| RIC online | 0.03 | **1.31** | 0.65 |
| PRO-WIC | **0.47** | 1.28 | **1.03** |

Table 5: Performance comparison across AlpacaEval 2, Arena-Hard, and MT-Bench benchmarks.

| Methods | AlpacaEval 2 | | Arena-Hard | | MT-Bench |
|---------|--------------|--------|------------|--------|----------|
| | WR(%) | LC(%) | WR(%) | SC(%) | Score |
| SFT | 34.03 | 34.08 | 48.5 | 44.3 | 7.71 |
| DPO | 37.24 | 36.84 | 49.0 | 47.2 | 7.83 |
| IPO | 37.95 | 36.43 | 54.6 | 48.3 | 7.64 |
| KTO | 38.12 | 36.51 | 43.9 | 44.1 | 7.63 |
| SIMPO | 40.03 | 40.78 | 54.6 | 48.8 | 7.58 |
| WPO | 44.11 | 40.06 | 62.0 | 53.0 | 7.81 |
| SELECTIVE DPO | 38.02 | 39.21 | 51.7 | 48.2 | 7.74 |
| PPO | 39.52 | 39.79 | 55.3 | 48.9 | 7.81 |
| ADPO | 44.04 | 38.90 | 61.9 | 53.2 | **7.97** |
| MORLHF | 41.38 | 44.83 | 44.2 | 34.1 | 7.20 |
| PRO-MORLHF | **47.30** | **50.35** | **63.5** | **54.2** | 7.93 |

context-specific weights for each prompt. The results demonstrate significant performance improvements across all benchmarks. These substantial improvements highlight the critical importance of prompt-aware preference adaptation.

## 6.4 EFFECT OF THE PREFERENCE ORCHESTRATOR

To further demonstrate the effectiveness of our *Preference Orchestrator*, we analyze the convergence behavior during training. Figure 3 shows the training reward curves for both our method PRO-MORLHF and the baseline MORLHF approach on the Ultrafeedback dataset.

As shown in the figure, the purple curve corresponds to PPO trained with a single reward model and improves slowly, whereas the other colored curves represent our PRO-MORLHF with an adapter that assigns prompt-specific weights; our method achieves much faster reward growth from early training and maintains a clear lead throughout, validating its efficiency and effectiveness.

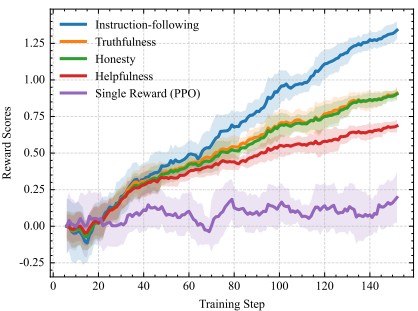

Figure 3: Training reward curves comparing PRO-MORLHF and PPO on Ultrafeedback dataset.

## 7 CONCLUSION

In this paper, we introduced the *Preference Orchestrator*, a novel approach for multi-objective alignment in large language models. By learning to predict context-specific preference weights based on input prompts, our method enables prompt-aware optimization that effectively balances multiple conflicting objectives. Theoretical analysis demonstrates that our approach achieves a smaller lower bound of alignment gap compared to fixed-weight methods. Extensive experiments on various datasets and benchmarks show that our method outperforms state-of-the-art baselines in both multi-objective alignment and general capability assessments.

ETHICS STATEMENT

We adhere to the ICLR Code of Ethics in this research, The datasets used are publicly available with no inclusion of private, sensitive, or proprietary data involving human/animal subjects. We are committed to ensuring that our research has a positive impact on society and the environment. We have conducted a thorough review of our research and its potential impact, and we have identified no significant ethical concerns.

REPRODUCIBILITY STATEMENT

Our work prioritizes reproducibility. All details for data preprocessing, model training, and evaluation are included in Appendix. The datasets used are all publicly accessible, and we have cited their corresponding literature in the paper.

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

# A APPENDIX

## A.1 PROOF OF THEOREM 5.1

We first relate the alignment gap to the difference between policies, then connect the policy difference to the difference in reward functions, and finally link the reward difference to the difference in weight vectors.

Firstly, we consider the alignment gap for a generic policy $\pi_{\boldsymbol{w}}$ that is optimal for a given weight vector $\boldsymbol{w}$. For a specific prompt $\boldsymbol{x}$, its gap with respect to the optimal policy $\pi^*_{\boldsymbol{w}^*(\boldsymbol{x})}$ is:

$$\text{Gap}(\pi_{\boldsymbol{w}}, \boldsymbol{x}) = F_{r_{\text{mo}}(\cdot; \boldsymbol{w}^*(\boldsymbol{x}))}(\pi^*_{\boldsymbol{w}^*(\boldsymbol{x})}) - F_{r_{\text{mo}}(\cdot; \boldsymbol{w}^*(\boldsymbol{x}))}(\pi_{\boldsymbol{w}}). \tag{15}$$

**Step 1: From Alignment Gap to Policy Difference.** By Assumption (i), the objective function $F_r(\pi)$ is $\mu$-strongly concave. This means that for any two policies $\pi_1, \pi_2$ and reward function $r$, we have:

$$F_r(\pi_1) - F_r(\pi_2) \geq \langle \nabla F_r(\pi_2), \pi_1 - \pi_2 \rangle + \frac{\mu}{2} \|\pi_1 - \pi_2\|^2. \tag{16}$$

Since $\pi^*_{\boldsymbol{w}^*(\boldsymbol{x})}$ is the maximizer of $F_{r_{\text{mo}}(\cdot; \boldsymbol{w}^*(\boldsymbol{x}))}(\cdot)$, the gradient at the optimum is zero, i.e., $\nabla F_{r_{\text{mo}}(\cdot; \boldsymbol{w}^*(\boldsymbol{x}))}(\pi^*_{\boldsymbol{w}^*(\boldsymbol{x})}) = 0$. Setting $\pi_1 = \pi^*_{\boldsymbol{w}^*(\boldsymbol{x})}$ and $\pi_2 = \pi_{\boldsymbol{w}}$, we get a lower bound on the gap:

$$\text{Gap}(\pi_{\boldsymbol{w}}, \boldsymbol{x}) \geq \frac{\mu}{2} \|\pi^*_{\boldsymbol{w}^*(\boldsymbol{x})} - \pi_{\boldsymbol{w}}\|^2, \tag{17}$$

where $\|\cdot\|$ denotes the norm in the policy space. Now utilizing that $\log \pi(\boldsymbol{y}|\boldsymbol{x})$ is Lipschitz continuous with parameter $L_\pi = \frac{1}{c}$, with the condition that there is some constant $c > 0$ such that $\pi(\boldsymbol{y}|\boldsymbol{x}) \geq c$ for all $\boldsymbol{x}, \boldsymbol{y}$, we have:

$$\| \log \pi^*_{\boldsymbol{w}^*(\boldsymbol{x})} - \log \pi_{\boldsymbol{w}} \| \leq L_\pi \|\pi^*_{\boldsymbol{w}^*(\boldsymbol{x})} - \pi_{\boldsymbol{w}}\|. \tag{18}$$

**Step 2: From Policy Difference to Reward Difference.** As shown in Direct Preference Optimization (DPO) (Rafailov et al., 2023), the optimal policy for the KL-regularized objective has an analytical form:

$$\pi_{\boldsymbol{w}}(\boldsymbol{y}|\boldsymbol{x}) = \frac{1}{Z(\boldsymbol{x})} \pi_{\text{ref}}(\boldsymbol{y}|\boldsymbol{x}) \exp\left(\frac{1}{\beta} r_{\text{mo}}(\boldsymbol{x}, \boldsymbol{y}; \boldsymbol{w})\right), \tag{19}$$

where $Z(\boldsymbol{x}, \boldsymbol{w})$ is a normalization constant. Taking the logarithm, we have:

$$\log \pi_{\boldsymbol{w}}(\boldsymbol{y}|\boldsymbol{x}) = \log \pi_{\text{ref}}(\boldsymbol{y}|\boldsymbol{x}) - \log Z(\boldsymbol{x}) + \frac{1}{\beta} r_{\text{mo}}(\boldsymbol{x}, \boldsymbol{y}; \boldsymbol{w}). \tag{20}$$

The difference in log-probabilities between two optimal policies is directly proportional to the difference in their corresponding reward functions:

$$\log \pi^*_{\boldsymbol{w}^*(\boldsymbol{x})} - \log \pi_{\boldsymbol{w}} = \frac{1}{\beta} \left( r_{\text{mo}}(\cdot; \boldsymbol{w}^*(\boldsymbol{x})) - r_{\text{mo}}(\cdot; \boldsymbol{w}) \right). \tag{21}$$

Combining this with Eq. 18, we get:

$$\text{Gap}(\pi_{\boldsymbol{w}}, \boldsymbol{x}) \geq \frac{\mu}{2 L_\pi^2 \beta^2} \| r_{\text{mo}}(\cdot; \boldsymbol{w}^*(\boldsymbol{x})) - r_{\text{mo}}(\cdot; \boldsymbol{w}) \|^2. \tag{22}$$

**Step 3: From Reward Difference to Weight Difference.** Now, we use Assumption (ii), the $L_r$-Bi-Lipschitz continuity of the reward function with respect to the weight vector $\boldsymbol{w}$. This implies:

$$\| r_{\text{mo}}(\cdot; \boldsymbol{w}^*(\boldsymbol{x})) - r_{\text{mo}}(\cdot; \boldsymbol{w}) \| \geq L_r \|\boldsymbol{w}^*(\boldsymbol{x}) - \boldsymbol{w}\|_2. \tag{23}$$

Squaring both sides and substituting into Eq. 22, we obtain a lower bound for the gap at a single prompt $\boldsymbol{x}$:

$$\text{Gap}(\pi_{\boldsymbol{w}}, \boldsymbol{x}) \geq \frac{\mu L_r^2}{2 \beta^2 L_\pi^2} \|\boldsymbol{w}^*(\boldsymbol{x}) - \boldsymbol{w}\|_2^2. \tag{24}$$

We can now apply this general result to our two specific policies, $\pi_{\text{fixed}}$ and $\pi_{\text{adapt}}$, by taking the expectation over the prompt distribution $\mathcal{D}$.

Table 6: Hyperparameters for Qwen2.5-7B during generation and training.

| Hyperparameters | Notation | Qwen2.5-7B |
|---|---|---|
| *Generation* | | |
| Temperature | - | 0.8 |
| Top-p | - | 0.95 |
| Generation Num | K | 8 |
| Max_new_token | $L_{\text{new}}$ | 2048 |
| *Training* | | |
| Learning rate | $\alpha$ | 5e-7 |
| Batch size | B | 128 |
| Max prompt length | $L_{\text{prompt}}$ | 2048 |
| Max generation length | $L_{\text{gen}}$ | 2048 |
| Training max length | $L_{\text{max}}$ | 4096 |
| Reward model max length | $L_{\text{reward}}$ | 4096 |
| KL loss | $\beta$ | 0.1 (2.5 for SimPO) |

For the fixed-weight policy, $\pi_{\text{fixed}}$, the weight vector is always $\boldsymbol{w} = \boldsymbol{w}_{\text{fixed}}$. Taking the expectation of Eq. 24 over $\boldsymbol{x} \sim \mathcal{D}$:

$$\text{Align-Gap}(\pi_{\text{fixed}}) = \mathbb{E}_{\boldsymbol{x} \sim \mathcal{D}}[\text{Gap}(\pi_{\text{fixed}}, \boldsymbol{x})] \geq \frac{\mu L_r^2}{2\beta^2 L_\pi^2} \mathbb{E}_{\boldsymbol{x} \sim \mathcal{D}} \left[ \|\boldsymbol{w}^*(\boldsymbol{x}) - \boldsymbol{w}_{\text{fixed}}\|_2^2 \right]. \quad (25)$$

For $\pi_{\text{adapt}}$, the weight vector for each prompt $\boldsymbol{x}$ is given by our preference orchestrator, $\boldsymbol{w} = f_\psi(\boldsymbol{x})$. Taking the expectation of Eq. 24 over $\boldsymbol{x} \sim \mathcal{D}$:

$$\text{Align-Gap}(\pi_{\text{adapt}}) = \mathbb{E}_{\boldsymbol{x} \sim \mathcal{D}}[\text{Gap}(\pi_{\text{adapt}}, \boldsymbol{x})] \geq \frac{L_r^2}{2\mu\beta^2} \mathbb{E}_{\boldsymbol{x} \sim \mathcal{D}} \left[ \|\boldsymbol{w}^*(\boldsymbol{x}) - f_\psi(\boldsymbol{x})\|_2^2 \right]. \quad (26)$$

By the theory of generalization error bound with assumption (iii) (Mohri et al., 2018; Liu et al., 2023; Xu et al., 2023), we have with probability at least $1 - \delta$,

$$\mathbb{E}_{\boldsymbol{x} \sim \mathcal{D}} \left[ \|\boldsymbol{w}^*(\boldsymbol{x}) - f_\psi(\boldsymbol{x})\|_2^2 \right] = C^2 \mathcal{O}(\frac{\log \frac{1}{\delta}}{N}). \quad (27)$$

Then, the proof is completed.

## A.2 IMPLEMENTATION DETAILS

We provide the implementation details of baselines and our method in the following subsections. In the test-time multi-objective alignment setting, we follow the implementation of RIC (Yang et al., 2024b). The backbone of the *Preference Orchestrator* is xlm-roberta-base [1]. We train the *Preference Orchestrator* with learning rate of 1e-5 and batch size of 32. The optimizer is AdamW and the temperature parameter $\tau$ is set to 0.1. For the PRO-WIC, the training step of offline stage is 10000 and the training step of online stage is 5000 for 2 epochs, in each epoch, we sample 5000 prompts from the prompt set for online sampling.

In the multi-objective alignment setting, we set the hyperparameters for baselines used in the experiments as listed in Table 6. For the PRO-MORLHF, we use the same backbone xlm-roberta-base and train the *Preference Orchestrator* with learning rate of 1e-5 and batch size of 32. The optimizer is AdamW and the temperature parameter $\tau$ is set to 0.1. All of the reward models are trained with the backbone of qwen2.5-0.5b [2]. Specifically, for the baselines that using single reward model, we train

---

[1] https://huggingface.co/FacebookAI/xlm-roberta-base
[2] https://huggingface.co/Qwen/Qwen2.5-0.5B

the reward model on the Ultrafeedback of the binarized version [3]. And for the methods that using multiple reward models, we sampled the preference pairs for each objective and train the reward model on the Ultrafeedback of the fine-grained version [4].

All experiments are conducted on 8 NVIDIA A800 GPUs and Intel(R) Xeon(R) Platinum 8358 CPU.

## B THE USE OF LARGE LANGUAGE MODELS

We acknowledge the use of a large language model (LLM) as an assistive tool in the preparation of this manuscript. The LLM's role was primarily confined to language refinement, including grammar and spelling checks, and enhancing the logical coherence and clarity of the prose. Additionally, the model assisted in the generation of certain segments of code. The core conceptual framework, theoretical analysis, experimental design, and conclusions presented in this paper are the original work of the authors.

---

[3]`https://huggingface.co/datasets/HuggingFaceH4/ultrafeedback_binarized`
[4]`https://huggingface.co/datasets/openbmb/UltraFeedback`

