# OpenReview forum: "Preference Orchestrator: Prompt-Aware Multi-Objective Alignment for Large Language Models"
_ICLR.cc/2026/Conference — ICLR 2026 Conference Withdrawn Submission_

### Official Review · Reviewer_nm6C · 2025-10-22

**Soundness:** 2
**Presentation:** 3
**Contribution:** 2
**Rating:** 2
**Confidence:** 3

**Summary:**

This paper introduces Preference Orchestrator, a new framework for multi-objective alignment in Large Language Models.  PRO uses a lightweight adapter that automatically infers the optimal preference weighting for each prompt, both during training and inference. The adapter is trained using normalized reward scores from multiple reward models on human-preferred responses, automatically capturing context-specific, effective preference trade-offs.

**Strengths:**

1. PRO automates the inference of preference weights for multi-objective alignment, removing reliance on manually specified or randomly sampled weights.
2. PRO is implemented as a flexible module that can enhance existing multi-objective alignment methods without significant changes to the base pipeline.
3. The paper mathematically proves that adaptive, prompt-aware preference weighting reduces the alignment gap compared to fixed-weight approaches, providing strong justification.

**Weaknesses:**

1. PRO assumes high-quality reward models are available for all objectives, which may not always be practical, especially for new domains or complex human values.
2. Theorem 5.1 hold only under the three stated assumptions. Are these three assumptions typically satisfied in practice? It is not intuitive to compare the two bounds (uniform weights and adaptive weights), since the bound of uniform weights depends on the optimal weights and the bound of adaptive weights does not.
3. The experiments lack comparison with more recent multi-objective alignment methods such as modpo and cpo.
4. There are some typos in the paper, e.g. "The is then trained using supervised learning to predict these implicit preference weights:"

**Questions:**

1. How the quality of reward models impact the performance of PRO? How to decide whether an off-the-shelf reward model can be used?
2. Equation (7) is confusing. Why a simple application of Softmax function is used here to derive the training target, without considering different weights of these reward models?

---

### Official Review · Reviewer_wCSK · 2025-10-27

**Soundness:** 3
**Presentation:** 3
**Contribution:** 2
**Rating:** 2
**Confidence:** 4

**Summary:**

This paper proposes Preference Orchestrator (PRO), a framework for prompt-aware multi-objective alignment of large language models (LLMs). Traditional RLHF or multi-reward alignment methods combine multiple reward models (e.g., helpfulness, harmlessness, truthfulness) using fixed scalar weights, which fail to capture how different prompts may emphasize different alignment dimensions. PRO introduces an orchestrator network that dynamically predicts reward weights based on the prompt, enabling adaptive combination of multiple reward signals during alignment.

The orchestrator $f_{\psi}(x)$ is a lightweight neural network that maps each prompt $x$ to a weight vector $w(x) = [w_1, \dots, w_K]$, representing the relative importance of $K$ reward models.
To train it, the authors construct pseudo-labels $w^\ast(x)$ by taking the softmax-normalized reward scores of the preferred responses under existing reward models.
The orchestrator is trained to minimize a KL divergence between $f_{\psi}(x)$ and $w^\ast(x)$ to infer prompt-specific reward priorities.

During policy optimization, the total reward becomes a dynamic weighted sum
$$
r_{\text{total}}(x, y) = \sum_k w_k(x) r_{\phi_k}(x, y),
$$
so that each prompt’s training signal reflects its inferred preference structure.

Experiments on summarization, reasoning, and dialogue datasets show that PRO improves multi-objective alignment compared to fixed-weight baselines, and a theoretical analysis suggests adaptive weighting yields a tighter lower bound on the alignment gap than static weighting.

**Strengths:**

1. The paper is original in reframing multi-objective alignment as a prompt-conditioned weighting problem. Instead of manually fixing or tuning scalar weights across objectives, the proposed Preference Orchestrator (PRO) introduces a neural adapter that dynamically infers the relative importance of each reward model based on the input prompt. This context-aware weighting is conceptually novel and provides a simple, modular mechanism that can integrate with RLHF pipelines.

2. The technical formulation is coherent and easy to follow. The orchestrator’s training via KL divergence to softmax-normalized pseudo-labels is straightforward, and the adaptive reward combination integrates seamlessly into existing policy optimization frameworks. Theoretical analysis provides an intuitive justification and the experimental setup is systematic across multiple benchmarks (summarization, reasoning, and dialogue).

3. The clarity of exposition is strong: figures and equations are cleanly presented, and the motivation is well articulated. The paper successfully communicates the high-level intuition behind dynamic reward balancing and includes sufficient implementation detail to enable replication of the main experiments.

4. In terms of significance, PRO offers an interpretable and computationally simple mechanism for handling heterogeneous alignment goals in LLMs. Although its conceptual novelty lies more in formulation than algorithmic depth, the paper contributes a useful perspective on how prompt-level conditioning can adapt multi-reward objectives, making it relevant for future explorations in adaptive and context-sensitive alignment strategies.

**Weaknesses:**

1. The main weakness of the paper lies in its methodological grounding and evaluation fairness. The proposed Preference Orchestrator (PRO) learns pseudo-label weights $w^{\ast}$ directly from softmax-normalized reward model outputs on preferred responses. This assumes that these pseudo-labels accurately reflect the true multi-objective preference structure, but such an assumption is tenuous—reward models themselves are noisy, biased, and not explicitly calibrated for cross-objective comparison. Consequently, PRO may simply learn correlations already embedded in the reward models, rather than genuine prompt-specific trade-offs.

2. Furthermore, the method is not compared to recent implicit-weight approaches such as Maxmin RLHF, Pareto-DPO, or dynamic multi-objective optimization frameworks that inherently adjust weighting during training. These approaches achieve adaptive trade-offs without relying on pseudo-supervision. The lack of comparison to such baselines makes it difficult to evaluate whether PRO provides any substantial benefit or theoretical advancement beyond heuristic reweighting.

3. In addition, the orchestrator may unintentionally cause reward specialization rather than true multi-objective alignment: for prompts strongly associated with a single objective, the softmax-derived pseudo-label tends to become one-hot, reinforcing only that dimension. This undermines the original goal of achieving balanced, simultaneous alignment across objectives.

**Questions:**

1. Comparison with Implicit-Weight Methods: Have the authors compared PRO with recent implicit multi-objective alignment frameworks such as Maxmin-RLHF: Alignment with Diverse Human Preferences (Chakraborty et al.) or other dynamic weighting approaches that optimize directly for trade-offs without learning explicit weights? These methods achieve adaptive balancing in a theoretically grounded way. Including such baselines would help determine whether PRO’s explicit orchestrator provides a genuine improvement in adaptability or merely reparameterizes existing implicit trade-offs.

2. The orchestrator is trained using pseudo-labels derived from softmax-normalized reward scores. How do the authors justify this as a reliable supervisory signal, given that reward models are not calibrated across objectives and may output scores on incomparable scales? More critically, this mechanism fails in conflicting-objective scenarios. For instance, in the helpful–harmless setup, a harmful prompt typically yields a very low harmlessness reward—even though harmlessness should dominate in such cases. Under PRO’s construction, the pseudo-label $w^\ast(x)$ would assign less weight to harmlessness precisely when it should be prioritized, meaning the orchestrator would reinforce the wrong trade-off. This suggests that pseudo-labels generated from raw reward magnitudes do not reliably represent the intended preference structure.

3. Reward Specialization (Critical) and Overfitting: Given that $w^\ast(x)$ could become one-hot for prompts strongly tied to a single objective, PRO may end up reinforcing reward specialization rather than achieving true multi-objective balance—particularly under conflicting objectives. This behavior fundamentally undermines the paper’s central claim of “simultaneous” alignment, as the model would effectively learn to optimize one reward per prompt rather than integrating multiple human preference dimensions. Could the authors analyze the entropy or smoothness of the learned weight distributions to verify whether this collapse occurs? Additionally, introducing regularization (e.g., entropy maximization, variance penalties, or cross-objective consistency constraints) might help enforce smoother and more balanced trade-offs across prompts.

4. Incomplete Sentence: The sentence on lines 237–238 appears incomplete.

---

### Official Review · Reviewer_M85v · 2025-10-28

**Soundness:** 2
**Presentation:** 2
**Contribution:** 2
**Rating:** 2
**Confidence:** 4

**Summary:**

This paper introduces Preference Orchestrator, which serves as a lightweight adapter that automatically assigns preference weights given an input prompt and can be integrated with existing multi-objective alignment methods. Overall, automatically determining preference weights is a reasonable research direction, but the current presentation and experiments should be further improved.

**Strengths:**

- This paper proposes a preference orchestrator that automatically determines the optimal preference weight vector for multi-objective
 alignment given an input prompt. This is important as we do need to find the optimal preference weight for each user prompt, helping us better construct the preference dataset or align LLMs in online RLHF settings.

- The overall writing is clear and easy for readers to follow.

**Weaknesses:**

Presentation:

- The main contribution of this paper is the proposal of the Preference Orchestrator. However, the key methodological details are missing (e.g., reward models, data source, and so on). Although the appendix provides some additional information, it still lacks sufficient information, and the author needs to include a reference to the appendix in the main paper.

- Lack of details. For example, what does MIC mean?

- The paper's central motivation (existing multi-objective alignment methods rely on manually specified preference weights) is not entirely correct. [1] uses expected preference values instead of preference weights. An example of the prompt used in CPO could be: [Helpfulness: 5, Harmlessness: 5], where the score of 5 is the expected reward instead of the preference weight. [2] does not use preference weight to align models.

Methodology:

- The current design simply uses reward models and applies softmax. This approach can be problematic. For example, for a prompt "How to make a cake", the reward of helpfulness and harmlessness can be both high (e.g., 5 score). However, it is clear that helpfulness should be prioritized.

- The rationale for using KL divergence is not explained.

Experiments:

- The choice of the XLM-RoBERTa-base model for the PRO is not justified. The use of a 0.5B parameter reward model seems suboptimal and limits performance.

- Since the PRO's training is dependent on the scores from a separate RM, an analysis of the PRO's robustness to the quality and biases of this underlying RM is essential and currently absent.

- Figure 2 and Table 1-2 compare against a very limited set of baselines. Comparisons against more recent SOTA multi-objective alignment methods are necessary.

- Table 5 claims to show advantages in multi-objective alignment, but the selected baselines are not specialized for multi-objective alignment.

- The ablation study comparing "PRO-WIC" and "RIC Variants" is flawed. Authors should compare:

PRO-WIC-Offline vs. RIC-Offline and PRO-WIC-Online vs. RIC-Online


References

[1] Controllable preference optimization: Toward controllable multi-objective alignment

[2] REWARD CONSISTENCY: Improving Multi-Objective Alignment from a Data-Centric Perspective

**Questions:**

How does the method perform if a user wants to specify custom preference weights?

---

### Official Review · Reviewer_8czu · 2025-10-31

**Soundness:** 2
**Presentation:** 3
**Contribution:** 3
**Rating:** 4
**Confidence:** 4

**Summary:**

The paper presents the challenge of LLM’s alignment: it is inherently hard to optimize multiple, frequently competing, objectives. The authors also claim that current multi-objective alignment approaches rely on given or manually defined preference weights, which present a burden on their users and result. Therefore, they introduce the Preference Orchestrator, which is a lightweight, prompt-aware adapter module.

More specifically, the core task of this adapter is to infer a context-specific weight vector for a provided input prompt. Formally, this adapter is defined as a neural network which aligns the prompt x to weight vector w on the probability simplex.  For this, the authors employ existing human preference datasets of the form $(x, y^+, y^-)$. Consequently, they assume that the reward profile of the preferred answer y^+ represents an implicit signal on the desired preference trade-off for the prompt x.

Since they have selected preferred responses, they determine a vector of scores of $K$ individual reward models. Next, they normalize this vector utilizing a softmax function with a set temperature parameter T and generate the target weight distribution (Equation 7). Then, the adapter is trained in a supervised manner to foresee this target distribution by minimizing KL-divergence between the output produced and the target.

Furthermore, the paper shows that PRO is a flexible plug-in module. It can be combined with ordinary Multi-Objective Reinforcement Learning from Human Feedback by dynamically supplying weights for the composite reward function. Similarly, they can utilize PRO to enhance test-time adaptive techniques by substituting random preference sampling with extra focused ones in the training curriculum as discussed.

**Strengths:**

1. They observe that users do fight to articulate the optimal balance of objectives for a certain query; in every case, a single fixed weighting is never the optimal one. It is inherently suboptimal across prompts.
2. The way to derive such signal from the reward scores of the existing preferred responses, as formulated in Equation 7 is a contribution. It transforms an ill-posed problem of discovering the “correct” preference weights into a standard supervised learning task and leverages current preference datasets even without requiring some new or special forms of human annotation. This will not only make the method more immediately applicable to plenty of existing alignment pipelines but reduce the barrier to adoption.
3. PRO can boost existing tools. Their demonstration on its immediate integration with both MORLHF in Equation 9 and test-time adaptive approaches (Section 4.3) is clear.
4. The author evaluates PRO in two significant and distinct settings: test-time adaptation and an entirely general capability assessment and conducts testing on multiple benchmarks such as AlpacaEval 2, MT-Bench, and Arena-Hard.

**Weaknesses:**

1. Imagine this case: humans may well prefer output $y^+$ to $y^-$ because it performs strictly better on a single criterion the user may particularly care about, as it is too important to get it wrong. In other words, the gap might be such that the learning system’s intention would vastly differ from the user’s. In that case, the softmax function would incorrectly perceive it as an equally dense weight vector and distort the user’s intent and teaching the adapter to pursue suboptimal trade-offs. How to solve this?
2. The method might be high sensitive to the quality and calibration of the underlying reward models. The training signal for the PRO adapter is a simple aggregation of the raw scores of K reward models as explained. Any malignancies in these reward models, including but not limited to biases, scaling issues, or inaccuracies, will be conveyed onto the architectural level (e.g., if one reward model is poorly calibrated and consistently outputs scores on a larger scale than the others, the adapter would learn to weigh this score more)
3. You state that the PRO adapter is “lightweight”. Have you conducted any empirical analysis regarding its computational overhead? What is its inference latency? How many parameters are added by the adapter to the overall model? What is the increase in total training time or memory when integrating PRO into an existing alignment pipeline?
4. The choice of baselines for the general capability assessment on the Ultrafeedback dataset (Table 5) might be inappropriate (or need to be supplemented). The proposed multi-objective method, PRO-MORLHF, is compared against a suite of methods that are designed for single-objective alignment (DPO, IPO, KTO, etc.). The paper notes that these baselines were trained on a binarized version of the dataset, while PRO-MORLHF leverages the fine-grained, multi-objective labels. A much fairer and more informative comparison would be against other state-of-the-art multi-objective alignment methods.
5. What is the proper choice of $T$ used in? My understanding of its implications is that low $T$ would lead to nearly one-hot target weights and steer the adapter to focus on a single objective, while high $T$ would lead the weights toward a bystander’s distribution. The appendix states that $T$ was set to 0.1, any ablation study or sensitivity analysis to justify this choice?

**Questions:**

1. Please refer to weakness 1.
2. Please refer to weakness 2. The training of the PRO adapter is conditioned on the output of the K reward models; some suggestion: you could try artificially degrading the reward of one of the reward models during adapter by adding a systematic bias or random noise to its outputs. Then, you should measure the impact on the final aligned model’s performance and whether this process affects the learned weight distributions?
3. Please refer to weakness 1 about “lightweight.” Some suggestions: (a) inference latency of the adapter module itself; (b) the parameter count vs. a base LLM, for instance, LLaMA-7B; and (c) the % increase in total training time when using the full PRO-MORLHF pipeline compared to a baseline MORLHF run for the same number of steps
4. Any explanation on weakness 4? some suggestion: some other established multi-objective alignment methods, such as MODPO (Zhou et al., 2024b) or CPO (Guo et al., 2024).
5. Please refer to weakness 5.

**Details Of Ethics Concerns:**

The prompt-aware preference adapter and its design as a modular plug-in are reasonable. On the other hand, it is highly sensitive to the quality of the underlying reward models. Additionally, more analyses are needed (see ‘review’)

---

### Note · Authors · 2026-01-06

I have read and agree with the venue's withdrawal policy on behalf of myself and my co-authors.